# An Up-to-Date Article Regarding Particularities of Drug Treatment in Patients with Chronic Heart Failure

**DOI:** 10.3390/jcm11072020

**Published:** 2022-04-04

**Authors:** Valentina Buda, Andreea Prelipcean, Dragos Cozma, Dana Emilia Man, Simona Negres, Alexandra Scurtu, Maria Suciu, Minodora Andor, Corina Danciu, Simina Crisan, Cristina Adriana Dehelean, Lucian Petrescu, Ciprian Rachieru

**Affiliations:** 1Faculty of Pharmacy, “Victor Babes” University of Medicine and Pharmacy, Eftimie Murgu Square, No. 2, 300041 Timisoara, Romania; buda.valentina@umft.ro (V.B.); andreea.preli@yahoo.com (A.P.); alexandra.scurtu@umft.ro (A.S.); suciu.maria@umft.ro (M.S.); corina.danciu@umft.ro (C.D.); cadehelean@umft.ro (C.A.D.); 2Research Center for Pharmaco-Toxicological Evaluation, “Victor Babes” University of Medicine and Pharmacy, Eftimie Murgu Square, No. 2, 300041 Timisoara, Romania; 3Faculty of Medicine, “Victor Babes” University of Medicine and Pharmacy, Eftimie Murgu Square, No. 2, 300041 Timisoara, Romania; man.dana@umft.ro (D.E.M.); andor.minodora@umft.ro (M.A.); simina.crisan@umft.ro (S.C.); petrescu_lucian@yahoo.com (L.P.); ciprian.rachieru@umft.ro (C.R.); 4Institute of Cardiovascular Diseases Timisoara, 13A Gheorghe Adam Street, 300310 Timisoara, Romania; 5Faculty of Pharmacy, “Carol Davila” University of Medicine and Pharmacy, Traian Vuia 6, 020956 Bucharest, Romania; simona_negres@yahoo.com; 6Center for Advanced Research in Cardiovascular Pathology and Hemostasis, “Victor Babes” University of Medicine and Pharmacy, Eftimie Murgu Square, No. 2, 300041 Timisoara, Romania

**Keywords:** heart failure, treatment strategies, new pharmacological approaches, age-induced changes, sex-related differences, pharmacokinetics, pharmacodynamics, discontinuation of treatment, food supplements, drug interactions

## Abstract

Since the prevalence of heart failure (HF) increases with age, HF is now one of the most common reasons for the hospitalization of elderly people. Although the treatment strategies and overall outcomes of HF patients have improved over time, hospitalization and mortality rates remain elevated, especially in developed countries where populations are aging. Therefore, this paper is intended to be a valuable multidisciplinary source of information for both doctors (cardiologists and general physicians) and pharmacists in order to decrease the morbidity and mortality of heart failure patients. We address several aspects regarding pharmacological treatment (including new approaches in HF treatment strategies [sacubitril/valsartan combination and sodium glucose co-transporter-2 inhibitors]), as well as the particularities of patients (age-induced changes and sex differences) and treatment (pharmacokinetic and pharmacodynamic changes in drugs; cardiorenal syndrome). The article also highlights several drugs and food supplements that may worsen the prognosis of HF patients and discusses some potential drug–drug interactions, their consequences and recommendations for health care providers, as well as the risks of adverse drug reactions and treatment discontinuation, as an interdisciplinary approach to treatment is essential for HF patients.

## 1. Introduction

Heart failure is defined by the European Society of Cardiology Guidelines as a clinical syndrome derived from structural and/or functional cardiac abnormalities [1]. This syndrome is characterized by common symptoms (such as fatigue, breathlessness, or ankle swelling) and typical signs (such as peripheral edema, elevated jugular venous pressure, or pulmonary crackles), all leading to reduced cardiac output and/or high intracardiac pressure (at rest or during stress periods) [2]. Thus, heart failure (HF) can be defined as the inability of the heart to ensure optimal blood flow, which is necessary for the organs to maintain the metabolic and functional processes of all organs. Currently, several HF definitions are available, which differ in the function of the setting (from the medical literature to medical practice, including the current guidelines) [3]. Moreover, several classification frameworks also exist, which aim to properly characterize different subsets of HF (from NYHA classification to EF categories or HF etiology) [3].

Drug therapy is a well-established strategy in treating heart failure (HF) and there are guidelines that cover most of the long-standing and recent research, although specific situations cannot be extensively analyzed [2].

HF affects a wide range of patients and thus occurs in several forms that are widely based on the assessment of left ventricular (LV) ejection fraction (EF): heart failure with preserved ejection fraction (HFpEF), with mildly reduced ejection fraction (HFmrEF), and with reduced ejection fraction (HFrEF) [4]. HFpEF is considered in patients with normal LVEF (commonly considered to be ≥50%, with symptoms and signs, elevated levels of natriuretic peptides and 1 additional criterion of relevant structural heart disease/diastolic dysfunction at least) [3]. The HFrEF designation is typically applied to patients presenting with less than 40% EF. Patients with LVEF between 40 and 49% represent a gray area, nowadays referred to as HFmrEF [5].

The diagnosis of HFpEF is relatively demanding, as patients usually have a normal LV, with a certain degree of wall hypertrophy and/or increased left atrial (LA) volume. Diagnosis requires proof of increased LV filling pressure or impaired LV filling, which explains the terminology of diastolic HF/dysfunction. However, diastolic dysfunction is also found in most HFrEF and HFmrEF patients (previously referred to as systolic HF) [2].

It seems that the prevalence of HF significantly varies with age, starting from approximately 1–2% among adults and increasing strikingly to more than 10% in people older than 70 years, and is more common in men than in women [2,6]. Since HF incidence and prevalence increase with age, HF is nowadays one of the most common reasons for hospitalization of elderly people [7]. Although treatment strategies and overall outcomes of HF patients have improved over time, hospitalization and mortality rates still remain elevated, especially in developed countries where the population is aging, which represents an economic burden for healthcare budgets [7].

Except for the pathology itself, HF is also associated with co-morbidities such as prior stroke or myocardial infarction, atrial fibrillation, diabetes mellitus, chronic lung disease, osteoarthritis, thyroid disease, dementia, depression, and chronic renal/hepatic failure, and all of these pathologies require additional treatment strategies [8].

Moreover, several aspects, such as changes in the mechanisms (neuroendocrine, inflammatory, immunological, or metabolic) involved in the physiopathogenesis of HF [7], the presence of co-morbidities, the use of polypharmacy in HF patients, and altered pharmacokinetics and pharmacodynamics of drugs in elderly people [9,10], require a patient-centered approach in order to avoid inappropriate medical prescriptions, drug interactions, exacerbation of adverse drug effects [11], and low adherence to pharmacological treatment, with altered prognosis of HF patients being a major consequence [12,13,14].

Moreover, because the worldwide population is aging and the number of people ≥80 years old will triple by 2050, it is extremely important to decrease the prevalence and incidence of cardiovascular pathologies in order to decrease multi-morbidity and health care costs [15].

Herein, we discuss the main important aspects regarding pharmacological approaches, treatment strategies, and the particularities of patients and treatment that should absolutely be taken into account in order to improve treatment outcomes for HF patients.

## 2. General Considerations Regarding HF Treatment

The pharmacological treatment of HF is oriented towards the following: long-term management of the pathology and improvement in survival (e.g., ACEIs (angiotensin-converting enzyme inhibitors), ARBs (angiotensin receptor blockers), beta-blockers, MRA (mineralocorticoid receptor antagonists), ARNI (angiotensin receptor-neprilysin inhibitors), SGLT2 (sodium-glucose cotransporter-2) inhibitors and ivabradine, an I_f_ channel blocker, highly selective for sinoatrial node pacemaker current) and symptom relief medication (e.g., administration of diuretics, nitrates or digoxin). Except for loop diuretics and digoxin, all of these options for treatment have been shown to improve symptoms, reduce hospitalization rates and/or prolong survival, in large randomized controlled trials [16,17].

Drug selection for HF depends on the type of HF and on the personal characteristics of the patients, the most important goals being as already mentioned: to reduce mortality, to improve clinical status and functional capacity and to prevent hospitalization [16,17].

### 2.1. Mechanisms of Action of the Classical Therapy in Chronic HF Patients

Later, we summarize the mechanisms of actions and the benefits of the main classes of drugs/pharmaceutical substances used in the treatment of HF.

Beta-blockers bind to β adrenergic receptors (β_1_-receptors located in the heart and kidneys; β_2_-receptors located in the vessels, lungs, gastrointestinal tract, liver, uterus, and skeletal muscle; β_3_-receptors located in the adipocytes). By binding toβ_1_-receptors, they block the deleterious actions of catecholamines: noradrenaline and adrenaline [18]. As a result, the heart rate and the contractility decrease and thus, the cardiac output and blood pressure will also decrease. As the heart rate will decrease, this will allow a longer time for diastolic filling, without typically reducing the stroke volume. Moreover, certain beta-blockers (cardioselective ones) will also reduce rennin secretion (via the blockade of β_1_ receptors in the juxtaglomerular apparatus), thus decreasing the severity of angiotensin II-induced vasoconstriction and aldosterone-induced volume expansion [19]. They are classified into noncardioselective beta-blockers (e.g., propranolol, carvedilol, and labetalol) and cardioselective beta-blockers (β_1_-selective, e.g., atenolol, metoprolol, bisoprolol and nebivolol). Certain beta-blockers are associated with vasodilating properties (nebivolol improves nitric oxide release, whereas carvedilol and labetalol block the α_1_-receptor). The vasodilating properties are beneficial because they decrease the peripheral vascular resistance, thus improving stroke volume, left ventricular function and therefore, cardiac output [18].

ACEIs selectively inhibit the angiotensin-converting enzyme leading to decreased angiotensin II production and, therefore, limit its negative effects, such as vasoconstriction, antidiuretic hormones, and aldosterone secretion. Moreover, ACEIs will increase the levels of the potent vasoactive peptide bradykinin, an endogenous vasodilator. Thus, ACEIs will induce vasodilatation, decreasing the total peripheral resistance (both arterial and venous) and blood pressure. In this way, they decrease the left ventricular afterload, thus increasing cardiac output and decreasing filling pressures (both left and right), which will improve pulmonary and systemic venous congestion [20].

ARBs work on the same angiotensin pathway, the difference being the fact that they bind to AT1 receptors located on the vascular smooth muscle, as well as in other tissues (e.g., heart) and thus, they block the damaging actions of angiotensin II. They induce less vasoconstriction and antidiuretic hormone and aldosterone secretion and lower blood pressure. Therefore, as well as ACEIs, they prevent damage to the vasculature, heart and kidneys [20].

As in some cases, the ACEIs or ARBs do not suppress the excessive formation of aldosterone sufficiently, patients with moderate to severe heart failure can also benefit from aldosterone antagonists. MRAs work by competitively blocking the binding of aldosterone to the mineralocorticoid receptor, thus decreasing the reabsorption of sodium and water, as well as decreasing the excretion of potassium, leading to cardioprotective effects [21].

Loop diuretics act by inhibiting the luminal sodium-potassium chloride cotransporter located in the thick ascending limb of the loop of Henle, where approximately 20–30% of the filtration of sodium occurs. Therefore, compared with other diuretics, loop diuretics reduce the reabsorption of a much greater proportion of sodium, leading to the excretion of it, alongside water. This will decrease the plasma volume, cardiac workload, and oxygen demand, thereby relieving the signs and symptoms of volume excess. They are currently used to relieve symptoms associated with pulmonary congestion and peripheral edema in HF patients [22].

If the patient is intolerant to ACEIs/ARBs/ARNIs, other vasodilators can be used, such as isosorbide dinitrate or hydralazine. Isosorbide dinitrate acts by releasing nitric oxide into the vascular smooth muscle cell, which activates guanylyl cyclase (an enzyme that catalyzes the formation of cyclic guanosine monophosphate-cGMP from guanosine triphosphate–GTP). Therefore, the increased intracellular cGMP will activate a series of reactions, which will decrease the intracellular calcium and thus, the contractility of vascular smooth muscle, leading to smooth muscle relaxation and vasodilatation. Hydralazine also acts on the vascular smooth muscle, with multiple effects such as the stimulation of nitric oxide release from the vascular endothelium (with cGMP production and low-intracellular calcium concentration), opening of potassium channels and inhibition of calcium release from the sarcoplasmic reticulum, thus inducing smooth muscle relaxation and subsequent vasodilatation [23].

Digoxin increases cardiac muscle cells’ contractility by inhibiting Na^+^/K^+^/ATPaze pump in the cardiac muscle, a pump responsible for moving sodium ions out of the cells and bringing potassium ions into the cells. When sodium concentrations in the cardiac cell increases, another electrolyte mover known as the sodium-calcium exchanger pushes the excess of the sodium ions out, while bringing additional calcium ions in. Therefore, the intracellular calcium increases, which will later increase the force of contraction and thus the cardiac output. Cardiac output increases followed by a decrease in ventricular filling pressures. Moreover, it inhibits the atrio-ventricular node, by stimulating the parasympathetic nervous system. Therefore, it diminishes the electrical conduction in the AV node and thus the heart rate. However, it has not been shown to reduce mortality [24].

Ivabradine acts by blocking the I_f_ current channel, responsible for the cardiac peacemaker, which regulates the heart rate. In this way, it prolongs the diastolic time and decreases the heart rate without affecting myocardial contraction/relaxation or ventricular repolarization [25].

### 2.2. New Approaches in HF Pharmacological Treatment

As several pharmacological classes of drugs have emerged in recent years with proven long-term benefits, in the following, we describe some of the most important aspects, as they are currently underused.

#### 2.2.1. Sacubitril/Valsartan

The combination of sacubitril and valsartan is the first from the class of angiotensin receptor–neprilysin inhibitors (ARNI). Agents in this new therapeutic class (sacubitril/valsartan) act at the level of RAAS and the neutral endopeptidase system. Sacubitril acts by inhibiting neprilysin and slowing down the degradation of natriuretic peptides, bradykinin, adrenomedullin, and other peptides [26]. It is indicated in chronic symptomatic heart failure with reduced ejection fraction [27].

Sacubitril/valsartan also improves symptom severity and heart functionality in patients with HFpEF, reducing the serum levels of the biomarker NT-pro BNP (and increasing BNP), an indicator of heart failure severity, and improves quality of life after 24 weeks [28].

One of the largest HF trials ever performed (PARADIGM-HF trial) compared enalapril with sacubitril/valsartan. In this trial, 8442 patients with HFrEF with FEVS ≤ 40% were enrolled and randomly received enalapril or sacubitril/valsartan twice daily. The trial was stopped early after 27 months because sacubitril/valsartan met the pre-specified stopping endpoint for an overwhelming benefit. All of the outcomes showed a 20% lower event rate in favor of sacubitril/valsartan; even the death rate from any cause was 16% lower in the group receiving sacubitril/valsartan [29]. ARNIs have been associated with improvements in diastolic function, left ventricular function, quality of life and decrease in ventricular arrythmias [30,31].

In the PROVE-HF and EVALUATE-HF trials, sacubitril/valsartan showed efficacy in improving the structural and functional changes that occur during heart failure. It improves cardiac remodeling and decreases the biomarker NT-pro BNP, so the drug reverses the damage to the heart in HFrEF patients [32].

Sacubitril/valsartan is recommended to replace ACE inhibitors when HFrEF patients are still symptomatic after optimal therapy. When initiating therapy with sacubitril/valsartan, there are some safety issues, including symptomatic hypotension, angioedema, and risk of hyperkalemia, so monitoring blood pressure levels, kidney function, and kalemia is extremely important [27,33]. Although the new combination was approved for the market starting from 2015, it is currently still underused, despite its proven benefits [34].

Figure 1 presents the mechanism of action of sacubitril/valsartan association and its consequences [27,28,29,30,31,32,33,34].

#### 2.2.2. Sodium Glucose Co-Transporter-2 Inhibitors

It is known that patients with type 2 diabetes mellitus (T2DM) are prone to developing cardiovascular events and heart failure, which can lead to high rates of hospitalization and premature mortality [35].

A new class of antidiabetics, sodium glucose co-transporter-2 (SGLT2) inhibitors, has also been found to have beneficial effects in patients with cardiac diseases [36,37]. The compounds in this class are represented by empagliflozin, dapagliflozin, canagliflozin, and ertugliflozin [38]. They act by inhibiting glucose transport in the proximal tube of the kidney, resulting in glucosuria and, as a result, lower blood glucose levels [35].

Aside from the direct mechanism of action on glucose control, other indirect mechanisms are taken into account regarding possible cardiovascular benefits [39].

In Figure 2, we summarize the possible mechanisms involved, their actions, and their effect on the heart [39,40].

The main trials reporting the benefits of SGLT2 inhibitors in HF patients with reduced EF, more precisely of dapagliflozin and empagliflozin, are as follows: DAPA-HF [41], DEFINE-HF [42] and EMPEROR-reduced [43].

The DAPA-HF trial evaluated the long-term effects of dapagliflozin on the incidence of cardiovascular death or HF hospitalization, regardless of the presence of diabetes. It was a phase 3 randomized placebo-controlled study, enrolling 4744 patients suffering from chronic HFrEF (NYHA class II-IV, LVEF ≤ 40% in addition to the recommended HF therapy, NT-proBNP high and eGFR ≥ 30 mL/min/1.73 m^2^) and having a median period of 18 months. The obtained results were as follows: a reduction in all-cause mortality and HF symptom aggravation, and the improvement in physical condition and overall quality of life. Their excellent benefits were seen very soon after starting the treatment with dapagliflozin. Regarding the incidence of adverse effects, they were attributed to volume depletion, renal dysfunction or hypoglycemia, but they did not differ between the studied groups [41].

The DEFINE-HF trial assessed the effect of dapagliflozin on the symptoms and biomarker plasmatic concentration of HFrEF patients (NYHA class II-III, LVEF ≤ 40%, eGFR ≥ 30 mL/min/1.73 m^2^ and with elevated natriuretic peptides). In total, 263 patients were included (taking either dapagliflozin 10 mg/once, daily, or a placebo, for a period of 12 weeks, in addition to the recommended HF therapy). Dapagliflozin induced an improvement in the patients’ health conditions or in their natriuretic peptides’ plasmatic concentrations [42].

The EMPEROR-reduced clinical trial evaluated the outcome of empagliflozin in patients with chronic HFrEF (NYHA class II-IV, LVEF ≤ 40%, eGFR ≥ 20 mL/min/1.73 m^2^). It was a double-blind clinical trial involving 3730 patients who received either empagliflozin (10 mg/once daily) or a placebo, in addition to the recommended HF therapy, for a median period of 18 months. Cardiovascular death and hospitalization rates (due to the worsening of HF) were reduced by empagliflozin, regardless of the presence of diabetes mellitus. The annual decline in the renal filtration rate was reduced, as well as the severity of renal complications. Non-complicated fungal infections of the genital tract were reported more often in patients taking empagliflozin [43].

Therefore, both substances are included as recommend treatments for HFrEF patients by the American and European guidelines [1,17].

The EMPEROR-preserved study assessed the effects of empagliflozin in patients with chronic HFpEF (NYHA class II-IV, LVEF ≥ 40%, eGFR ≥ 20 mL/min/1.73 m^2^). In total, 5988 patients were included, who were randomized 1:1 and received either empagliflozin (10 mg/once daily) or a placebo, in addition to their classical HF therapy. Over a period of 26.2 months, the primary outcome was obtained (decreased risk of hospitalization in HF patients, regardless of the presence/absence of diabetes). Beneficial effects were also seen in eGFR, without considering the renal outcomes by themselves. It is important to note the fact that the most used medicines for HFrEF have not shown benefits in patients with HFpEF; therefore, empagliflozin is superior in improving HF outcomes even in patients with HFpEF, which are symptomatic and stable [44,45].

In Table 1, we summarize the indications, contra-indications and cautions worth considering for ARNI and SGLT2 inhibitors [17].

## 3. Treatment Strategies in HF Patients

For the treatment of heart failure with preserved ejection fraction or with mildly reduced ejection fraction (LVEF ≥ 50% or LVEF between 40 and 49), the guidelines recommend the prescription of diuretics, as first line therapy [1]. The other drugs (ACEI or ARB, beta-blockers or MRA) may be considered as a second alternative [1].

The treatment strategy also focuses on treating co-morbidities such as: hypertension, atrial fibrillation, cardiac ischemic disease, pulmonary hypertension, diabetes mellitus, chronic kidney disease, COPD (chronic obstructive pulmonary disease), anemia and obesity. The optimal management of co-morbidities has been shown to improve symptoms and to improve the patient’s quality of life [2].

In the case of congestion, diuretics will be very effective and will improve the symptomatology. There is proof that nebivolol, candesartan, digoxin and spironolactone might reduce hospitalization for patients with HFpEF in sinus rhythm [46]. Moreover, besides empagliflozin, none of other drugs consistently met their primary endpoint in the clinical trials that were performed, and none reduced mortality and morbidity [44,45].

For patients in atrial fibrillation, the prescription of an anticoagulant is very important for reducing thrombo-embolic events [47]. For the control of heart rate, the use of digoxin, beta-blockers or verapamil/diltiazem is recommended, targeting an optimal rate control between 60 and 100 bpm [48].

Amiodarone and non-dihydropyridine calcium-channel blockers (CCB) are able to reduce heart rate, but due to their adverse effects profile, they should be replaced, if possible. In the case of a fast ventricular rate and symptoms, it might be appropriate to consider AV node ablation, and if there are indications for ICD (implantable cardioverter-defibrillator), AV node ablation with the implantation of CRT-D (cardiac resynchronization therapy–defibrillator) might be preferred. The rhythm control strategy has not been shown to be superior to the rate control strategy. Urgent cardioversion is indicated if atrial fibrillation is life threatening [49].

Regarding HFrEF treatment, the evidence base for drug treatment in HF is for HFrEF. Either an ACEI/ARB/ARNI or a beta-blocker should be started (sometimes also ACEI/ARB/ARNI and beta-blocker at the same time), with doses up-titrated to the maximum tolerated/targeted dose every 2 weeks. ACEI, beta-blockers and MRA proved to improve survival and are recommended for the treatment of every patient with HFrEF. The new ARNI (sacubitril/valsartan) has been shown to be superior to ACEI in reducing the risk of death and hospitalization. Thus, ARNI is recommended to replace ACEI in cases of HFrEF patients if they are symptomatic despite optimal therapy [26].

In the case of decompensated patients, beta-blockers should not be initiated or if already initiated but patients develop worsening of HF symptoms (e.g., fatigue, dyspnea, dizziness or erectile dysfunction) caution should be applied regarding their prescription. Moreover, in the case of frailty or other complications (e.g., marginal hemodynamics), a longer period of time may be required for dose up-titration [17].

ARNI can be prescribed as an alternative to ACEI/ARB intolerance (e.g., angioedema) or in the absence of hypotension, electrolyte or renal imbalance. It is recommended to avoid the association of an ARNI with an ACEI and if previously administered ACEI, to ensure a 36 h washout period before the initiation of an ARNI, due to the high risk of angioedema [50]. This delay is not required when switching from ARB to ARNI. When up-titrating ARNI/ACEI/ARB (every 2 weeks or more), the monitoring of the potassium level, renal function and blood pressure is required. Lower loop diuretic doses may be necessary for the optimal titration of ARNI/ACEI/ARB and caution regarding the potassium concentration is required, as well as the dietary restriction of/supplementation with potassium, as the kaliuretic effect of loop diuretics might no longer be present [17].

If the patients have LVEF ≤ 35%, the guidelines recommend the use of MRAs to reduce mortality and hospitalization. MRAs (e.g., spironolactone or eplerenone) are added in patients with symptomatic chronic HFrEF, as a triple therapy (ACEI/ARB/ARNI + beta-blockers + MRA), in the absence of contra-indications. It is essential to achieve the targeted dose of other drugs before initiating the treatment with an aldosterone antagonist and to monitor the potassium levels and renal function under the treatment [17].

SGLT2 inhibitors can also be added, as part of the quadruple therapy (ACEI/ARB/ARNI + beta-blocker + MRA + SGLT2 inhibitor), in the absence of contra-indications. There is no need to achieve targeted doses of other drugs before adding SGLT2 inhibitors, although the loop diuretic dose might require adjustments based on the close monitoring of symptoms and weight [17].

Isosorbide dinitrate/Hydralazine could be prescribed especially for African American patients once the targeted dose of ACEI/ARB/ARNI + beta-blockers + MRA has been achieved [17].

The I_f_ channel inhibitor ivabradine is recommended in patients with symptomatic HFrEF or LVEF ≤ 35%, in sinus rhythm and heart rate ≥ 70 bpm, and in patients that have been hospitalized for HF in the last year, despite receiving beta-blockers at the maximum tolerated dose, ACEI and an MRA. The titration of the dose should be performed every 2 weeks in order to decrease the heart rate. In the case of patients ≥ 75 years old or in those with a history of conduction defects, the recommended initial dose is 2.5 mg twice daily, administered with meals [17].

## 4. Particularities of Patients

### 4.1. Age-Induced Changes

#### 4.1.1. Cardiovascular Structure and Function

A reduction in the response after beta adrenergic stimulation was observed (due to impaired coupling of G-protein receptors to adenyl cyclase and a decrease in adenyl cyclase concentration), which damages the capacity of the aging heart to increase cAMP as a response to the stimulation of beta receptors [7,51]. Thus, age-related cardiovascular changes are associated with a reduction in chronotropic and inotropic responses, which decline with age (peak contractility and heart rate decline almost linearly with age) [52].

The filling of left ventricular diastole is impaired by the aging process, as it is a process that depends on energy and active myocardial relaxation. Altered calcium release by the cardiomyocytes, with resulting prolonged contractile period of the heart, was also observed in elderly people [7,53].

The high deposits of collagen, amyloid, and lipofuscin in the interstitial space and myocyte hypertrophy seen in older people increase cardiac stiffness and decrease cardiac compliance, altering cardiac filling, especially in critical situations [7,54,55,56].

The increased vascular (arterial) stiffness (due to collagen deposition and cross-linking in the vascular media and to fragmentation of arterial elastin), together with impaired endothelium-dependent vasorelaxation (a consequence of vascular inflammation and altered endothelial nitric oxide synthesis) observed in aging lead to a higher afterload and a predisposition to systolic hypertension in the elderly [57,58].

Inadequate mitochondrial synthesis of adenosine triphosphate in response to stress will lead to altered energy release, thus altered cellular reactions, such as gene expression, chromatin remodeling, intra/extra-cellular signaling, ion homeostasis, muscle contraction, protein and hormone synthesis and secretion, and neurotransmitter release and reuptake [59].

#### 4.1.2. Other Organs

Age-associated modifications in the glomerular filtration rate and electrolyte imbalances [60,61] often seen in the elderly (due to dehydration, diuretic use, etc.) can raise the risk of HF decompensation and exacerbate the risk of drug side effects, with dangerous consequences, especially if the patient also has chronic kidney disease [62].

The aging of the respiratory system can lead to decreased compliance in pulmonary function. Moreover, the presence of chronic lung disease or sleep-related breathing disorders can increase the risk of pulmonary hypertension, exacerbate the sensation of dyspnea, and decrease biventricular filling [7,63,64].

The aging of the autonomic nervous system is characterized by sympathetic hyperreactivity and increased plasma concentrations of catecholamines, but a reduced sympathetic response is observed due to the diminished response of catecholamine receptors. Thus, tachycardia is felt less in elderly than middle-aged adults [65].

### 4.2. Sex Differences in HF

Regarding sex differences in heart failure, it seems that a large percentage of women tend to develop HRpEF, with the etiology of HF being either hypertension, diastolic dysfunction, or valvular pathology, whereas men tend to develop HFrEF or HFmrEF (HF with mid-range ejection fraction), with the etiology usually being an ischemic condition [66]. Moreover, it seems that women with HF are usually older and present with increased EF and more frequent symptoms linked to HF. Although they tend to also have multiple comorbidities compared to men, a meta-analysis showed that they have a better prognostic rate regarding hospitalization and mortality risk, regardless of EF [67].

The cardioprotection found in women seems to be due to the secretion of 17β-estradiol, an estrogen with a very clear established role in counteracting ischemic, hypertrophic, apoptotic, and cytotoxic impulses related to the heart [66,68,69].

Animal studies have shown that the cardiomyocytes of female models had a higher rate of survival after they were exposed to oxidative stress, which led to cell death, with the explanation relying on the fact that highly expressed estrogen receptor alpha (ER-α) can mediate the inhibition of pro-apoptotic pathways and the activation of the Akt signaling pathway [66,70].

Other differences regarding plasma B-type natriuretic peptide (BNP) levels, left ventricular mass index, left ventricular ejection fraction, and peak oxygen consumption between the sexes were also noted, suggesting that men are more susceptible to HF development than women [66,71,72].

Concerning treatment, it seems that although angiotensin-converting enzyme (ACE) inhibitors decrease the morbidity and mortality rates in both men and women, their effect seems to be more pronounced in men [73]. On the contrary, angiotensin II receptor blockers (ARBs) seem to have a higher mortality reduction rate in women than in men, although no difference was observed between the two classes of drugs (ACE inhibitors and ARBs) in terms of reduced mortality rates [74]. All of these aspects could be due to the action of estrogen on the receptor expression of angiotensin II by the ACE2 gene, located on chromosome X, and to the higher incidence of coughing and thus higher rate of discontinuation of ACE inhibitors in women [75].

No sex-related differences were observed in terms of treatment outcomes in patients under treatment with beta blockers or mineralocorticoid receptor antagonist [66].

Differences between the sexes were also noted in the case of digoxin treatment; women with a digoxin plasma concentration of 1.2–2.0 ng/mL had a higher mortality rate than men, although plasma levels of 0.5–0.9 ng/mL in men were associated with reduced mortality, but not any effect in women [76].

## 5. Particularities of Treatment

### 5.1. Pharmacokinetic Considerations and Their Consequences in HF Patients

Reduced blood flow to the gastrointestinal tract causes decreased absorption of drugs [77]. In the case of medicines with low permeability into the intestinal tissue, edema in the intestinal mucosa may affect their transport into the intestine [13,78].

Intestinal wall dysfunction secondary to hypoperfusion can, over time, induce chronic enteral inflammation and malnutrition. On the other hand, increased intestinal permeability in patients with HF can stimulate the transfer of drugs from the gastrointestinal tract to the portal blood [13].

Decreased blood perfusion in the central and peripheral organs results in an irregular tissue distribution of drugs [79]. Differences in the body’s water load can also affect the distribution of drugs [80,81].

Plasma protein binding may also be affected, especially after a myocardial infarction (production of α1-glycoproteins in the liver increases tissue necrosis and inflammatory reactions in the myocardium) or in patients with cachexia [82].

Reduced hepatic and renal blood flow induces an altered metabolism and elimination of administered drugs and their metabolites. In addition to an irregular distribution of drugs in the liver (a consequence of poor hepatic blood infusion), hepatic congestion and/or hypoxia (as a major consequence) and hepatocellular lesions may occur, manifested by hepatocytolysis (and thus increased liver transaminases) and disorders affecting enzymatic activity [83,84].

Since the concentration of active substances at the site of action cannot yet be directly determined, plasma concentration is often measured as a surrogate marker of the drug effect, depending on the concentration of active substances at the site of action [80].

Practically, changes in pharmacokinetics have been only observed in patients with renal and/or hepatic complications [13,84,85].

Increased action of the following drugs was observed after oral administration in patients with decompensated HF: captopril, enalapril, perindopril, carvedilol, felodipine, candesartan, furosemide, milrinone, and enoximone [13,84,85].

Since most studies to date (clinical trials) have not included patients with decompensated HF or major renal or hepatic problems (which involve more severe changes in PK and PD), the pharmacokinetic and pharmacodynamic parameters are currently under-studied in patients with HF. Thus, we recommend paying more attention to monitoring the efficacy and safety of drugs used in HF. Furthermore, the progressive titration of drugs should be implemented and the benefit/risk ratio should be periodically evaluated [13,84,85].

### 5.2. Pharmadynamic Considerations

The pharmacodynamics of drugs, as well as their tolerability, may also be affected by several neuronal and endocrinological compensatory mechanisms in HF, including the activation of the renin-angiotensin (RAA) and sympathetic system. Moreover, nodal activity and baroreceptor sensitivity are affected, and peripheral vascular resistance is increased; these are aspects that could cause an altered response to administered drugs [13,84,85].

The activation of the sympathetic nervous system can alter the perfusion of the viscera, especially the splenic organs (liver, gastrointestinal tract, kidneys), to maintain the perfusion of vital organs (brain and heart), resulting in hypoperfusion in the liver and kidneys. Furthermore, increased central pressure in patients with right HF causes hepatic congestion and dilation of the central vein in the hepatic acini, inducing hepatocellular ischemia and necrosis, and reducing the activity of microsomal enzymes [13,84,85,86].

Therefore, it is advisable to consider all changes that might occur in heart failure patients (Figure 3) [13,84,85,86].

### 5.3. HF Treatment in Patients with Cardiorenal Syndrome

It is well known that the acute/chronic dysfunction of one organ could induce the acute/chronic dysfunction of the other organ [87], therefore, cardiorenal syndrome has been defined as a spectrum of diseases involving the heart and the kidneys. This syndrome implies a “hemodynamic cross-talk” between the injured heart and the kidneys’ responses and vice versa [87,88]. Several mechanisms underline this cardiorenal syndrome, such as the hemodynamic interactions between the heart and kidneys in HF patients; cytokine production; the impact of atherosclerotic disease on both organs; biochemical perturbations due to the installation of chronic kidney disease; and the structural changes that appear in the heart, which are due to kidney disease progression [87,89].

In summary, the drop in cardiac output induces the activation of the sympathetic nervous system which will increase the stroke volume and the heart rate, as a compensatory mechanism. Sympathetic nervous system activation will also stimulate the release of renin from the kidneys, with the consequence of RAAS activation. Moreover, the drop in cardiac output will also induce a decreased perfusion of the kidneys, leading to kidney injuries (the beginning of cardiorenal syndrome). A reduced perfusion in the kidneys will stimulate renin release, RAAS activation and thus sodium and water retention (due to aldosterone secretion and antidiuretic hormone release), which will later increase the mean arterial pressure and the preload and decrease the cardiac output. RAAS activation will also cause vasoconstriction, which will contribute to reduced renal perfusion [87,89,90].

Moreover, chronic kidney disease (CKD) can lead to cardiovascular dysfunction, as a low glomerular filtration rate activates RAAS, which will lead, in time, to cardiac remodeling and left ventricular hypertrophy. CKD also implies a reduction in erythropoietin production over time, leading to anemia, which will increase the risk of ischemic events in the heart. Moreover, CKD induces a decrease in vitamin D production and parathormone stimulation, leading to an increase in calcium and phosphate levels and thus, increased risk of coronary and vessel calcification, augmenting the high risk of ischemic events [91]. Electrolyte imbalances are also observed in CKD patients, more precisely, hyperkalemia, which can increase the risk of cardiovascular complications [87].

Therefore, the management of cardiorenal syndrome is challenging and must be directed towards the specific pathophysiologic mechanism involved. The volume overload can be either addressed by prescribing diuretics (usually loop diuretics, as they are the most potent diuretics e.g., furosemide, torsemide and bumetanide) or using ultrafiltration methods. The addition of a thiazide diuretic to a loop diuretic may be preferred in the case of diuretic resistance, as an initial approach to restore euvolemia [87,89]. Regarding HF treatment in patients with renal disease, the renal function and potassium level should be checked within 1–2 weeks of the initiation or up-titration of an ACEI/ARB/ARN. Regarding aldosterone antagonists (MRA), in patients with preserved renal function or mild to moderate impairment, potassium levels and renal function should be checked within 2–3 days after the initiation of the therapy, followed by a check after 7 days of treatment, and at least monthly for the first 3 months and then, every 3 months [17].

In patients with severe renal impairment (eGFR < 30 mL/min/1.73 m^2^), the ARBs/ACEIs are considered safe. The starting dose of ARNI (Sacubitril/Valsartan) should be reduced to 24/26 mg, twice a day, in patients with severe renal impairment. The dose of ARNI might also need to be reduced in the case of hypotension or hyperkalemia. MRAs are contraindicated in patients with severe renal impairment, creatinine > 2.5 mg/dL in men, creatinine > 2 mg/dL in women or potassium > 5.0 mEq/L. As for SGLT2 inhibitors, there is currently no evidence regarding dose adjustments in patients with eGFR < 30 mL/min/1.73 m^2^ for dapagliflozin and eGFR < 20 mL/min/1.73 m^2^ for empagliflozin [17]. As a general rule, a decrease in eGFR of more than 30% or the apparition of hyperkalemia should alert the clinician to adjust (decrease) the doses of HF drugs [17].

### 5.4. HF Treatment in Pregnancy and Lactation

During pregnancy, the increased physiological requirements are partially fulfilled through changes in the physiology of the cardiovascular system, which has to adapt to the extra metabolic demands of the fetus and of the other organ systems. Therefore, the augmentation of the size and activity of the uterus, as well as the increase in blood flow in the choriodecidual space, represents extra work for the cardiovascular system. Moreover, during pregnancy, the skin and kidneys have an increased perfusion which allows them to disperse heat and retain sodium and water [92,93].

Symptoms of HF are more likely to appear in the second trimester as a consequence of an increased cardiac output and of intravascular volume (during pregnancy, the plasma volume increases by 40% and the cardiac output by 30–50%) [94]. Therefore, the therapeutical management of HF during pregnancy will be adapted to the clinical setting and the severity of the pathology. For cases in which the oral administration of drugs is sufficient, diuretics, betablockers, hydralazine or nitrates can be recommended. Usually, diuretics represent the first line treatment for pregnant HF women due to the increased preload associated with pregnancy (therefore, reducing preload will diminish the left side filling pressure and the pulmonary capillary pressure, and thus, it will allow the resorption of the pulmonary interstitial fluid). Currently, there is no evidence that diuretics are directly responsible for fetal growth restrictions. Betablockers decrease the heart rate and allow a greater filling during diastole. Beta-1-selective blockers (for example metoprolol succinate or bisoprolol) are preferred and better tolerated [1,92].

Regarding the management of HF before pregnancy, ACEIs, ARBs, ARNI, MRA and SGLT2 inhibitors, as well as ivabradine should all be avoided and stopped prior to conception due to an increased risk of fetal harm. It is recommended that the pregnancy be planned and closely monitored by a multidisciplinary team of specialists in order to avoid HF decompensation and fetal harm (induced by either the pathology or the treatment) [1].

Hydralazine, methyldopa, or oral nitrates can also be recommended during pregnancy [1].

In patients with atrial fibrillation, low-molecular-weight heparins are the first choice of anticoagulant treatment (NOAC should be avoided due to insufficient data regarding their safety) [1].

Concerning the breastfeeding period in HF women, it is important to know that the mother’s treatment prevail over breastfeeding compatibility, and that the benefits of breastfeeding are important for both the mother and child [95,96]. Enalapril is among the preferred options of ACEIs (as it has the most assuring safety data) and can be used from birth. [95]. As a second option, among the ARBs, losartan can be a good choice, due to its extensive first-pass metabolism and thus low systemic concentration, but breastfeeding should be performed with caution [95]. From the betablockers, metoprolol succinate or propranolol are the preferred choices (favorable PK profile and assuring data). Additionally, carvedilol or bisoprolol can be seen as a second option of treatment. Sacubitril/valsartan association should be avoided due to lack of data regarding their use during pregnancy, as well as SGLT2 inhibitors. Moreover, there is good evidence for digoxin, hydralazine and spironolactone use during breastfeeding period. The monitorization of babies exposed to either betablockers or ACEIs is recommended for hypotension (especially in neonates), lethargy, drowsiness, bradycardia, poor feeding, or weight gain [95].

## 6. Drugs and Food Supplements That Can Aggravate HF

The treatment of HF patients is very complex and includes not only lifestyle changes but also multiple pharmacological therapies, as well as the presence of co-morbidities and individual pharmacological strategies; this leads to polypharmacy in HF patients, generating increased iatrogenic risks [97,98].

In Appendix A we summarize the main/most used drugs and food supplements that can worsen the prognosis of heart failure patients; thus, it is recommended to avoid them by this category of patients [99,100,101,102,103,104,105,106,107,108,109,110,111,112,113,114,115,116,117,118,119,120,121,122,123,124,125,126,127,128,129,130,131,132,133,134,135,136,137,138,139,140,141,142,143,144,145,146,147,148,149,150,151,152,153,154,155,156,157,158,159,160,161,162,163,164,165,166,167,168,169,170].

## 7. Potential Drug–Drug Interactions in HF Patients

Several studies have shown a strong association between the number of drugs taken by HF patients (usually more than five) and the occurrence of potential drug–drug interactions, leading to the conclusion that the incidence of drug–drug interactions in HF patients is extremely high [97,171,172].

The coordinated efforts of a multidisciplinary team of healthcare providers also involving a clinical pharmacist could reduce the medication-related problems and improve the efficacy, tolerability, and safety of the pharmacological strategies implemented by physicians [171,173,174].

Appendix B summarizes the main important drug–drug interactions that should be considered in HF patients, their consequences, and some recommendations regarding their management [174,175,176,177,178,179,180,181,182,183,184,185].

## 8. Adverse Drug Reactions in HF Patients

Adverse drug reactions (ADRs) have been estimated to account for approximatively 10–20% of hospital admissions in geriatric units [186]. Moreover, an observational study performed in 1996 highlighted that the iatrogenic problems accounted for nearly 7% of HF admissions and were associated with higher mortality and prolonged hospital stays compared with those of non-iatrogenic causes [187,188]. Thus, the decompensation of HF patients due to iatrogenic conditions is a well-known and documented problem, which leads to increased morbidity and mortality rates. Therefore, good management of all prescribed drugs is mandatory for HF patients.

It seems that there are also sex-related differences between men and women regarding ADRs. Although women are underrepresented in all phases of clinical trials and little is known about this aspect, several meta-analyses concluded that women are more susceptible (1.5–1.7×) to developing ADRs than men and are also at higher risk of hospitalization due to the severity of ADRs [189,190].

As women usually present HFpEF with additional risk factors (co-morbidities and advanced age) compared with other types of HF, there seems to be a high incidence of polypharmacy, as they tend to take more drugs than men (including over-the-counter drugs and food supplements); thus, they have an increased risk of iatrogenic events (due to ADRs and drug interactions) and low adherence to treatment. Other explanations may underline this problem of high iatrogenic risk. Sex differences in the pharmacokinetics and pharmacodynamics of administered drugs (regarding distribution volume, hepatic/renal clearance, sex hormones, alterations in drug target expression and signal transduction pathways, immunological conditions, etc.) predispose women to a higher probability of overdosing than men [190,191,192,193,194]. Drug-induced ventricular arrhythmia (torsade de pointes) is more often encountered in women, as women have longer QTc intervals, probably due to the sex hormone modulation of Ca^2+^ and K^+^ channels implicated in ventricular repolarization [190,195]. Differences in prescribing habits for men and women compared with the recommended guidelines is another reason supporting the high incidence of ADRs in women, as well as the overall poor quality of life observed in women HF patients [189,190].

All of the aforementioned sex-related differences in female patients predispose women to a higher probability of drug-induced complications such as bleeding problems (e.g., under antithrombotics), electrolyte abnormalities (e.g., under diuretics), cough and increased creatinine (under treatment with ACE inhibitors), myopathy (under statin treatment), hepatotoxicity, skin diseases, etc. [190,196,197].

Thus, it is important to adjust the drug dosage as a function of total body weight/size or glomerular filtration rate and titrate it to the required clinical effect, especially in those with a narrow therapeutic index, in order to avoid the incidence of ADRs [189,190].

## 9. Discontinuation of Drugs in HF Patients

Several articles also discuss the negative outcomes of HF patients after discontinuing chronic HF treatment [2,187,198,199,200].

It was observed that RAAS (renin-angiotensin-aldosterone system) inhibitors provide the most beneficial outcomes in terms of mortality reduction in patients with HFrEF, although renal function is affected at baseline [201]. The cessation of these drugs in patients with HFrEF was associated with increased mortality and re-hospitalization admissions after 1 month, 3 months, and 1 year, which led to the conclusion that RAAS inhibitors should not be discontinued in patients with moderate to several renal dysfunction if the benefits outweigh the risks [202].

Regarding beta blocker discontinuation, although they are associated with a risk of negative inotropic effects and hypotension, ESC has recommended not to disrupt beta blocker treatment unless severe hypotension is present, due to the risk of rebound effects (such as rebound tachycardia, aggravation of angina pectoris, risk of ventricular arrhythmia) and the correlation with increased mortality and readmissions rates after cessation of treatment [2,199,203]. Several trials highlighted that continuous administration of beta blockers in patients with decompensated HF reduced the mortality and readmission rates [203,204,205].

## 10. Conclusions

In order to reduce exacerbations, hospital readmission rates, morbidity, and mortality and to improve the overall quality of life, an interdisciplinary approach to treatment strategies is mandatory for HF patients. The treatment strategy must be individualized for each HF patient, periodically monitored and reviewed by the healthcare team. Moreover, patient education, including topics such as dietary counseling, healthy lifestyle habits, regular exercise in a tolerable amount, alcohol and smoking cessation; moreover, understanding the alarming signs and symptoms of HF decompensation (shortness of breath, fatigue, ankle swelling, sudden weight modification) is another extremely important action that needs to be urgently implemented by societies with aging populations.

## Figures and Tables

**Figure 1 jcm-11-02020-f001:**
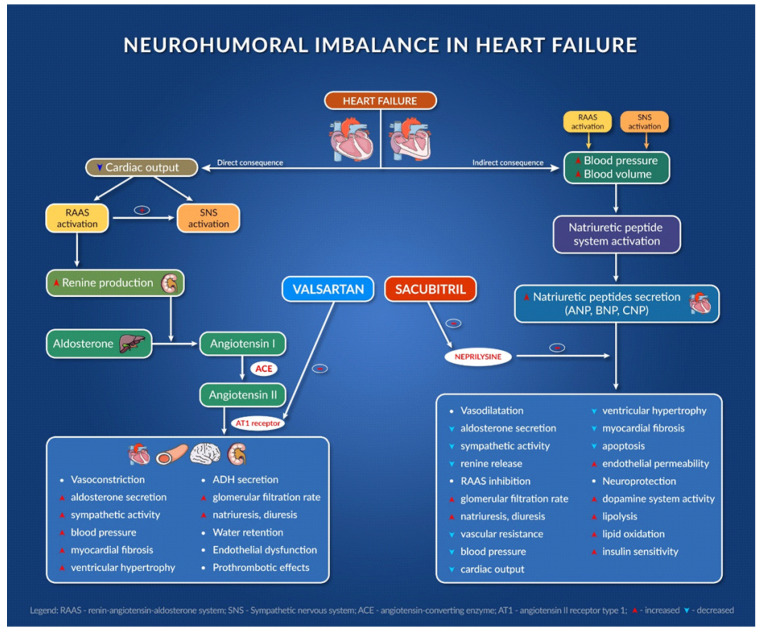
Neurohumoral imbalance in heart failure.

**Figure 2 jcm-11-02020-f002:**
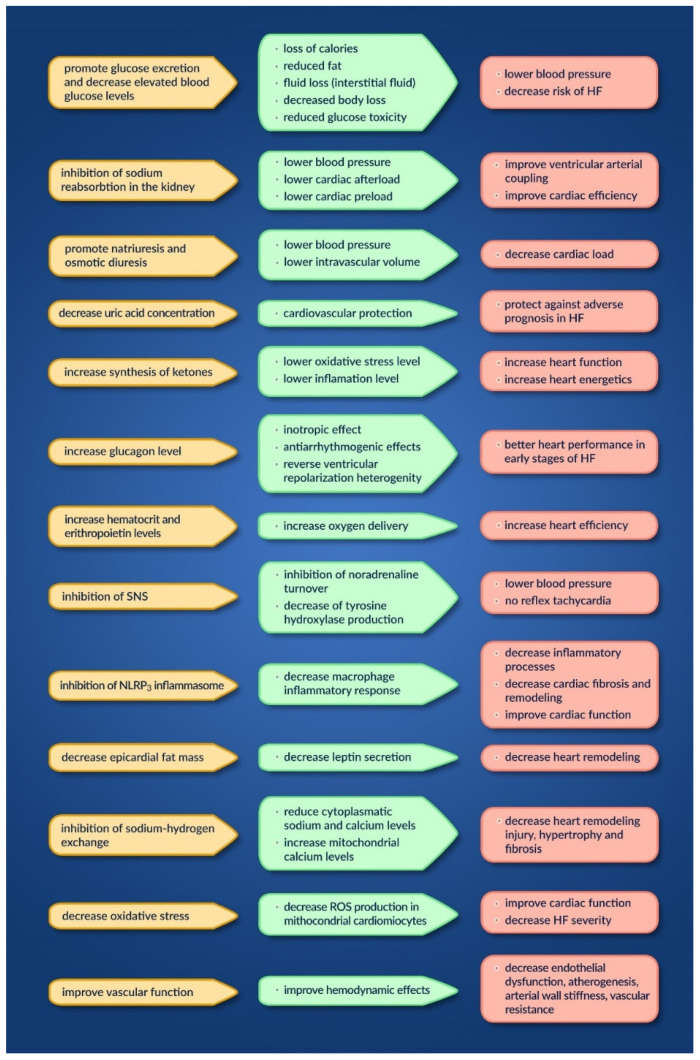
Mechanisms, actions and effects of SGLT2 inhibitors on the heart.

**Figure 3 jcm-11-02020-f003:**
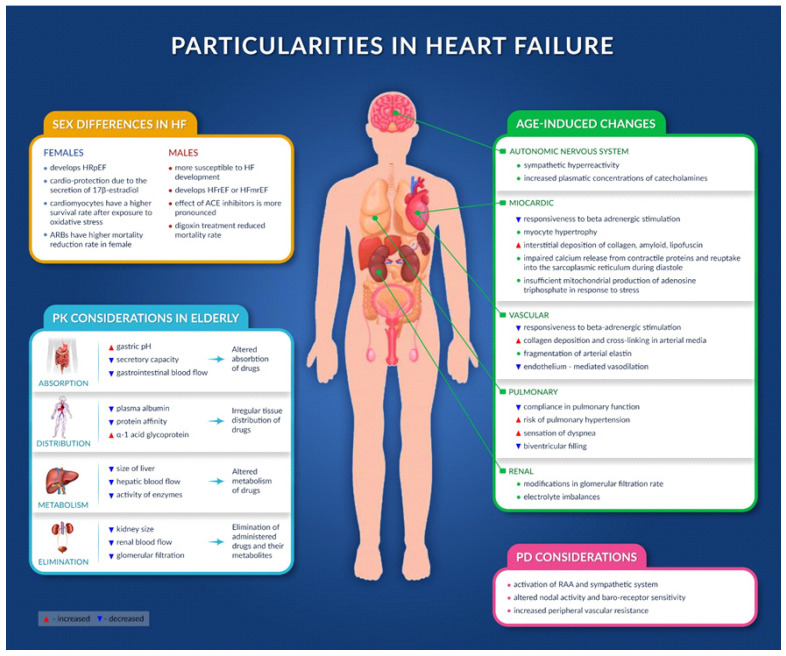
Particularities in heart failure.

**Table 1 jcm-11-02020-t001:** The indications, contra-indications and cautions for ARNI and SGLT2 inhibitors.

	ARNI	SGLT2 Inhibitors
Indications	▪ HFrEF (≤40%) ▪ NYHA class II-IV ▪ Alternative of ACEI/ARB	▪ HFrEF (≤40%) ± diabetes mellitus ▪ NYHA class II-IV
Contra-indications	- hypersensitivity to the active substances - history of angioedema - severe hepatic impairment - ≤36 h of the last ACEI dose	- hypersensitivity to the active substance - type I diabetes - dialysis - eGFR < 30 mL/min/1.73 m^2^ (dapagliflozin) - eGFR < 20 mL/min/1.73 m^2^ (empagliflozin)
Cautions	◊ severe renal impairment (starting dose: 24/26 mg × 2/day) ◊ moderate hepatic impairment (starting dose: 24/26 mg × 2/day) ◊ SBP < 100 mmHg ◊ volume depletion ◊ renal artery stenosis ◊ pregnancy/lactation	◊ high risk of genital infections (especially mycotic) and urinary infections ◊ hypovolemia ◊ ketoacidosis ◊ acute renal impairment ◊ necrotizing fasciitis of the perineum (Fournier gangrene) ◊ bladder cancer ◊ pregnancy

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
