# Peer review of "An Up-to-Date Article Regarding Particularities of Drug Treatment in Patients with Chronic Heart Failure"

_jcm, 2022, doi:10.3390/jcm11072020_

Round 1

Reviewer 1 Report

a review on the management of heart failure, Difficult to be concise while at the same time having all the relevant information. Few comments

In the introduction where you mention HFpEF, please mention the full diagnostic criteria as what you have written now is slightly misleading.

Section 2.1 betablockers, give examples of betablockers that reduce renin secretion

Page 4 spelling of hydralazine is wrong

Line 164 should be “decrease” not “decreased”

Line 271 should be “proofs”

Section 5.4 should be elaborated on especially the cardiorenal syndrome as deteriorating renal function is a major problem encountered in treating patients with heart failure

A section on pregnancy and lactation would also be beneficial

Appendix A title should be prognosis not prognostic

Author Response

Dear Peer-Reviewer,

In the name of all co-authors of the present article, we would like to thank you and the editorial staff for all the effort you put in reviewing the present manuscript. Your pertinent suggestions had increased its scientific value.

Please find enclosed our response to your comments.

Reviewer 1

A review on the management of heart failure, Difficult to be concise while at the same time having all the relevant information. Few comments

In the introduction where you mention HFpEF, please mention the full diagnostic criteria as what you have written now is slightly misleading.

Response: Thank you very much for the suggestion. We added the missing information (Lines 60 – 63)

Section 2.1 betablockers, give examples of betablockers that reduce renin secretion

Response: Thank you very much for the suggestion. More information was added regarding betablockers, including examples of cardioselective betablockers (Lines 118 - 134)

Page 4 spelling of hydralazine is wrong

Response: Thank you very much for the suggestion. We corrected the spelling of the word in the hole manuscript.

Line 164 should be “decrease” not “decreased”

            Response: Thank you very much for the suggestion. We corrected it.

Line 271 should be “proofs”

Response: Thank you very much for the suggestion. We corrected it.

Section 5.4 should be elaborated on especially the cardiorenal syndrome as deteriorating renal function is a major problem encountered in treating patients with heart failure

            Response: Thank you very much for the suggestion. It was very helpful. We added the missing information. Please see Lines 506 - 545

A section on pregnancy and lactation would also be beneficial

            Response: Thank you very much for the suggestion. It was very helpful. We added the missing information. Please see Lines 563 - 610

Appendix A title should be prognosis not prognostic

Response: Thank you very much for the suggestion. We corrected it. Moreover, the entire manuscript has been edited for English editing by MDPI Services.

Yours sincerely,

Dr. Dragos Cozma
Associate Professor
Institute of Cardiovascular Diseases, Timisoara

Faculty of Medicine
"Victor Babes" University of Medicine and Pharmacy Timisoara
Eftimie Murgu Square no.2, RO-300041

Reviewer 2 Report

The authors have attempted the difficult task of writing a comprehensive review of heart failure management. In my opinion, parts of the manuscript are too superficial, lack important references, and are not entirely in line with the latest guideline recommendations.

  • In the introduction, the authors should reference the current ESC heart failure guidelines and the Universal definition and classification of heart failure (doi: 10.1002/ejhf.2115).
  • HFmrEF should be Heart failure with mildly-reduced ejection fraction instead of mid-range ejection fraction.
  • Please add a reference to the following sentence: “Since HF incidence and prevalence increase with age, HF is nowadays one of the most common reasons for hospitalization of elderly people.”
  • Lien 110: not all beta-blockers are beta1-selective. Not only noradrenaline binds to beta1-receptors but also adrenaline.
  • Hidralazine should be Hydralazine.
  • Section 2.1 “Mechanisms of action of the classical therapy in chronic HF patients“ lacks references.
  • EMPEROR Preserved should be discussed.
  • The authors write that in patients with EF >40% the guidelines recommend ACEi/ARB, BB, MRA, and diuretics. This is not true. The guidelines state that only diuretics are recommended. The other drugs may be considered (Class IIB, Level C).
  • The authors write that “There are proves that nebivolol, candesartan, digoxin and spironolactone might reduce hospitalization for patients with HFpEF in sinus rhythm.” However, the authors should underline, that besides empagliflozine none of the drugs has consistently met their primary endpoint in trials and none has reduced mortality and morbidity.

Author Response

Dear Peer-Reviewer,

In the name of all co-authors of the present article, we would like to thank you and the editorial staff for all the effort you put in reviewing the present manuscript. Your pertinent suggestions had increased its scientific value.

Please find enclosed our response to your comments.

Reviewer 2

The authors have attempted the difficult task of writing a comprehensive review of heart failure management. In my opinion, parts of the manuscript are too superficial, lack important references, and are not entirely in line with the latest guideline recommendations.

Response: Thank you very much for your comment. We tried to add more data and references to the present manuscript.

  • In the introduction, the authors should reference the current ESC heart failure guidelines and the Universal definition and classification of heart failure (doi: 10.1002/ejhf.2115).

Response: Thank you very much for the suggestion. We added the two references.

  • HFmrEF should be Heart failure with mildly-reduced ejection fraction instead of mid-range ejection fraction.

Response: Thank you very much for the suggestion. We corrected it. (Line 59)

  • Please add a reference to the following sentence: “Since HF incidence and prevalence increase with age, HF is nowadays one of the most common reasons for hospitalization of elderly people.”

Response: Thank you very much for the suggestion. We added it.

  • Lien 110: not all beta-blockers are beta1-selective. Not only noradrenaline binds to beta1-receptors but also adrenaline.

Response: Thank you very much for the suggestion. More information was added regarding betablockers, including adrenaline. (Lines 118 - 134)

  • Hidralazine should be Hydralazine.

Response: Thank you very much for the suggestion. We corrected the spelling of the word in the hole manuscript.

  • Section 2.1 “Mechanisms of action of the classical therapy in chronic HF patients“ lacks references.

Response: Thank you very much for the suggestion. We added it.

  • EMPEROR Preserved should be discussed.

Response: Thank you very much for the suggestion. It was very helpful. We added the missing information. Please see Lines 280 - 290

  • The authors write that in patients with EF >40% the guidelines recommend ACEi/ARB, BB, MRA, and diuretics. This is not true. The guidelines state that only diuretics are recommended. The other drugs may be considered (Class IIB, Level C).

Response: Thank you very much for the suggestion. You apologize for this error. We made the necessary correction. Please see Lines 298 - 300

  • The authors write that “There are proves that nebivolol, candesartan, digoxin and spironolactone might reduce hospitalization for patients with HFpEF in sinus rhythm.” However, the authors should underline, that besides empagliflozine none of the drugs has consistently met their primary endpoint in trials and none has reduced mortality and morbidity.

Response: Thank you very much for the suggestion. It was very helpful. We added you comment in the manuscript. Please see Lines 310 - 312

Moreover, the entire manuscript has been edited for English editing by MDPI Services.

Yours sincerely,

Dr. Dragos Cozma
Associate Professor
Institute of Cardiovascular Diseases, Timisoara

Faculty of Medicine
"Victor Babes" University of Medicine and Pharmacy Timisoara
Eftimie Murgu Square no.2, RO-300041

Round 2

Reviewer 1 Report

The authors have succesfully answered all my queries

Reviewer 2 Report

The authors addressed all my comments satisfactorily and revised the manuscript adequately.

This manuscript is a resubmission of an earlier submission. The following is a list of the peer review reports and author responses from that submission.